# Patterns and Functional Insights of DNA Methylation Variation in a South American Mayfly Across an Agriculturally Impacted Semi-Arid Watershed

**DOI:** 10.3390/biology15010090

**Published:** 2025-12-31

**Authors:** Angéline Bertin, Ana María Notte, Bouziane Moumen, Diana Coral-Santacruz, Frédéric Grandjean, Nicolas Gouin

**Affiliations:** 1Departamento de Biología, Facultad de Ciencias, Universidad de La Serena, Raúl Bitrán 1305, La Serena 1700000, Chile; anotte@userena.cl (A.M.N.); diana.coral@userena.cl (D.C.-S.); ngouin@userena.cl (N.G.); 2Instituto de Ecología y Biodiversidad (IEB), La Serena 1700000, Chile; 3Doctorado en Biología y Ecología Aplicada, Universidad de La Serena, La Serena 1700000, Chile; 4Laboratoire d’Écologie et Biologie des Interactions (EBI), UMR CNRS 7267, Université de Poitiers, 86073 Poitiers, France; bouziane.moumen@univ-poitiers.fr (B.M.); frederic.grandjean@univ-poitiers.fr (F.G.); 5Centro de Estudios Avanzados en Zonas Áridas (CEAZA), Raúl Bitrán 1305, La Serena 1700000, Chile

**Keywords:** DNA methylation, epigenetic variation, environmental epigenomics, environmental stress, freshwater ecosystems, semi-arid watersheds, insect

## Abstract

Climate change and human activities threaten aquatic organisms that are vital for the health and balance of rivers and streams. For protecting freshwater ecosystems, it is thus crucial to understand how these organisms respond and adapt to environmental stress. In this study, we aimed to characterize some of the molecular responses of a native Chilean mayfly, *Andesiops torrens*, across natural populations inhabiting a semi-arid watershed under strong agricultural pressures. We focused on DNA methylation, a natural process that can switch genes on or off. We found differences in the DNA methylation patterns of mayflies depending on their locations within the watershed. This result indicates that local environmental conditions drive molecular changes in this species. The detected responses were mainly linked to the regulation of vital biological functions and the management of environmental stress. Our results demonstrate that mayflies respond to stress through molecular mechanisms and that DNA methylation analysis can help understand the pressures faced by natural populations, which is crucial for their conservation and sustainable management.

## 1. Introduction

Accelerating environmental change imposes intense and fast-paced ecological and evolutionary pressures on most living organisms, demanding rapid phenotypic responses to ensure population persistence and ultimately prevent species extinction [1]. Although evolutionary theory has long predicted that genetic responses would be too slow to keep pace with current rates of environmental change, evidence of rapid adaptive phenotypic responses is accumulating across a wide range of species [2,3,4,5], suggesting a role for non-genetic mechanisms. Among these, epigenetic processes have attracted particular scientific interest for their potential to mediate rapid evolutionary responses [6,7,8], with growing evidence indicating that epigenetic variation is widespread in natural populations [9]. Epigenetic mechanisms enable organisms to maintain homeostasis by regulating gene expression for essential cellular functions, while also allowing for adaptation and change in response to environmental stimuli [10,11]. They operate through heritable and reversible modifications influencing gene activity without altering the underlying DNA sequence [10,11]. Epigenetic modifications can be triggered by environmental cues and promote phenotypic plasticity, enabling organisms to adjust rapidly to environmental fluctuations [8]. In addition, spontaneous epimutations, which result from random changes in molecular pathways affecting epigenetic marks [12,13], may serve as a source of variation subject to natural selection [1]. Because spontaneous epimutations occur at rates several orders of magnitude higher than DNA mutations, they can offer a faster and more flexible substrate for natural selection [14]. They can be particularly relevant in natural populations with limited genetic variation and even in clonal species [15,16]. By contributing to rapid acclimation and local adaptation, epimutations may thus drive adaptive phenotypic divergence among genetically similar populations [17,18,19].

In plants, animals, and microorganisms, DNA methylation has emerged as a central epigenetic mechanism of gene expression regulation [20,21,22]. It typically involves the addition of a methyl group to cytosine residues in different sequence contexts (CG, CHG, and CHH, where H refers to A, C, or T), although in animals DNA methylation predominantly occurs in the CG (CpG) context. While DNA methylation levels are generally high, broadly distributed, and often associated with gene silencing in vertebrates and plants, in arthropods, it most commonly occurs within the transcribed regions of genes and is associated with active gene expression [23,24].

DNA methylation plays crucial roles in development [25], genome stability, transposable element silencing [8], and responses to environmental stimuli [26]. Exposure to extreme temperatures, pollutants, salinity, and other stressors has been shown to induce important methylation changes across various taxa, correlating with changes in gene activity and phenotypic outcomes [5,18,27]. Thus, DNA methylation screening provides a promising approach to track environmental pressures [26], with potential applications for biodiversity conservation and the management of vulnerable populations. Nonetheless, the use of DNA methylation screening to identify environmental pressures remains limited in wild, non-model populations exposed to complex and intense anthropogenic pressures.

In this study, we investigate genome-wide DNA methylation patterns using the methylRAD method [28] in wild populations of the mayfly *Andesiops torrens* (Lugo-Ortiz & McCafferty, 1999) (Ephemeroptera: Baetidae) along the Limarí watershed of north-central Chile, a semi-arid agricultural basin facing increasingly severe environmental stress. The watershed is characterized by prolonged droughts, high solar radiation, and pronounced seasonal temperature fluctuations, with precipitation occurring almost exclusively during the winter months. In recent decades, persistent declines in precipitation have led to sustained decreases in river flow, a trend further amplified by water extraction for agricultural use. This situation is placing increasing stress on freshwater biota, as reduced river flow leads to higher water temperatures, lower oxygen availability, and elevated concentrations of contaminants. Agricultural expansion further intensifies these effects through increased pesticide use, leading to widespread contamination of freshwater ecosystems with multiple agrochemicals, including mainly triazines, carbamates, organophosphates, and pyrethroids, and sometimes forming potent mixtures [29]. Consequently, aquatic organisms are increasingly exposed to a combination of chemical and physical stressors, which frequently exhibit synergetic effects [30,31,32].

The endemic mayfly *A. torrens* is well-suited for investigating epigenetic responses to environmental stressors. Previous studies in the same system have revealed limited genetic differentiation among populations [33], suggesting that mechanisms other than genetic variation may mediate phenotypic responses to environmental variation along the Limarí watershed. This hypothesis is supported by recent evidence of significant differential methylation at six methylRAD loci between pesticide-exposed and unexposed populations, indicating that chemical stress can trigger genome-wide shifts in methylation patterns [34]. However, the general methylation structure among populations and its biological relevance remain unclear. We hypothesized that, despite low genetic differentiation, *A. torrens* populations exhibit spatial structuring of DNA methylation along the Limarí watershed, and that loci associated with cellular regulation and stress-related responses may contribute to this structure. To test this, we characterized genome-wide DNA methylation profiles across multiple *A. torrens* populations distributed along environmental gradients in the Limarí watershed and identified the most discriminating methylated loci. We then examined the functional relevance of these loci to uncover the biological processes most associated with methylation-based population differentiation. This study reveals DNA methylation structure at both inter- and intra-site scales, offering new insights into the epigenetic landscape of natural populations, and illustrates the potential of DNA methylation analyses to uncover environmentally responsive processes in freshwater ecosystems.

## 2. Methods

### 2.1. Study System

The mayfly *Andesiops torrens* (Ephemeroptera, Baetidae) is a freshwater grazer [35] endemic to Chile, distributed from the Limarí watershed in the north (30° S) to the southern tip of Patagonia (54° S) [36], and found at altitudes between 40 and 2000 metres [37,38]. It has a univoltine life cycle, with adult emergence occurring between August and October in central Chile [36]. *A. torrens* prefers good-quality running waters, typically headwater streams. This species is present in agricultural areas at the northern limit of its distribution [33], such as in the Limarí watershed. In this semi-arid region [39], freshwater species face challenging conditions due to droughts, high water demand for human activities, and contamination by agrichemicals [33,40,41]. In this context, given its sensitivity to water condition, *A. torrens* represents a valuable model for assessing the impacts of environmental stress on freshwater biodiversity, as demonstrated by recent studies revealing genetic and epigenetic responses to pesticide contamination in water [33,34].

Genomic data for Ephemeroptera remains scarce, with only four genomes published to date (*Cloeon dipterum*, *Neocloeon triangulifer*, *Ephemera danica*, *Siphlonurus alternatus*). Consistent with patterns reported across the order [23], our draft genome of *A. torrens* (see below) revealed the presence of the enzyme Dnmt1, the maintenance methyltransferase [42], but no identifiable Dnmt3. As *DNMT1* can also function as a de novo methyltransferase in insects [43], it may compensate for the evolutionary loss of *DNMT3* in insects.

### 2.2. Biological Sampling and MethylRAD Production

In this study, we used the methylRAD data produced by Gouin et al. [34]. Sampling collection has been fully described in Gouin et al. [33,34]. Briefly, *A. torrens* nymphs were collected at 30 sites along the Limarí watershed in north-central Chile between May and June 2015, covering a range of environmental and spatial conditions (Figure 1). At each site, 9–10 nymphs (Appendix A) were hand-collected using entomological forceps and preserved in 95% ethanol at 4 °C. All individuals were selected at comparable developmental stages to minimize ontogonetic variability in methylation profiles. Genomic DNA was extracted using a CTAB protocol [33], its integrity checked on 1% agarose gels, and was quantified with the Quant-iT PicoGreen dsDNA Assay (Thermo Fisher Scientific, Wilmington, NC, USA). MethylRAD libraries were constructed for 285 *A. torrens* specimens by Oebiotech Co., Ltd. (Qingdao, China) using 500 ng of DNA at a concentration of 10 ng/μL following the multi-isoRAD protocol [28]. Each sample was digested with the Mrr-like enzyme *Fsp*EI, which targets methylated cytosines and generates 32 bp fragments. Fifty-seven multiplexed libraries, each containing five randomly selected specimens, were sequenced as paired-end reads on an Illumina HiSeq X Ten platform across five lanes [34]. A total of 3.78 billion raw reads were generated, which were merged using PEAR software v0.9.6 [44], and their adaptor sequences were removed. After quality filtering (Phred quality ≥ 30 and <8% ambiguous bases), 758 million high-quality (HQ) reads were retained [34]. HQ reads were pooled to generate de novo reference sites with CD-HIT v4.6 [45], as described by Wang et al. [28]. The HQ reads from each sample were then mapped against these reference sites using SOAP2 [46], resulting in the detection of a total of 1,377,147 methylated sites (665,987 CCGG and 711,160 CCWGG), with 58% located within annotated genes (31% in exons and 27% in introns) and 42% in intergenic regions. For relative quantification of methylRAD data, DNA methylation levels were quantified using the normalized read depth (reads per million, RPM) at each site [28], calculated as RPM = (reads coverage per site/high quality reads per library) × 1,000,000.

### 2.3. Reference Genome

To generate the draft genome of *A. torrens*, we used a hybrid sequencing strategy combining long- and short-read whole-genome sequencing, using PacBio Sequel II and Illumina NovaSeq PE150 platforms, respectively. All individuals used for sequencing originated from a single site in the Aconcagua River basin (32.50° S, 70.58° W, WGS 84) sampled in August 2021, where *A. torrens* were abundant and large-bodied. Genomic DNA was extracted from whole-body tissue using the E.Z.N.A. Insect DNA kit (Omega Bio-Tek, Norcross, GA, USA), with protocol modifications to reduce the high protein and RNA content, involving extended incubation with the CTL buffer (2 h at 37 °C instead of 30 min at 60 °C) and prolonged RNAse treatment (1 h at 37 °C instead of 10 min at 70 °C). For short-read sequencing, DNA was extracted from a single individual, while long-read sequencing used pooled DNA from 14 individuals to reach required DNA amounts. Genome sequencing, de novo assembly, and annotation were carried out at Macrogen (Seoul, Republic of Korea). All samples passed quality control checks for library construction. For PacBio Sequel sequencing, 6 µg of genomic DNA was employed to construct a 10 kb library using PacBio SMRTbell Express Template Prep Kit 2.0. The library average insert size was 19,791 bp. SMRTbell templates were annealed using Sequel II Bind Kit 2.0 and Internal Control 1.0, and then sequencing was performed using the Sequel II Sequencing Kit 2.0 and cells (8M Tray, with 15 h movies). For Illumina NovaSeq sequencing, the library was prepared from 1 µg of genomic DNA using the TruSeq PCR-free DNA High Throughput Library Prep Kit (Illumina), following the manufacturer’s instructions. The library average insert size was 470 bp. Paired-end sequencing was performed using one lane of a NovaSeq 6000 S4 flow cell to produce approximately 110 Gb of data.

Subreads generated from PacBio Sequel II were assembled using Wtdbg2 v2.3 [47], with a genome size option of 145 Mb and a coverage cutoff of 100×. Assembly results were polished using the Arrow algorithm of GCpp 2.0.0 (https://github.com/PacificBiosciences/gcpp, accessed on 8 December 2025). Then, the Illumina NovaSeq raw reads were filtered by quality, keeping only reads with 90% of bases with a Phred score of 30 or higher for error correction. Adapters were trimmed from the filtered reads with Trimmomatic 0.38 [48] using the following option (ILLUMINACLIP:Adapter.fasta:2:30:10:8:true LEADING:15 TRAILING:15 SLIDINGWINDOW:4:15 MINLEN:36). The assembly from the PacBio reads was corrected with Pilon v1.21 [49] three times using these high-quality NovaSeq reads. SNAP v2.31.8 [50] was used to make a gene training model with proteins and CDS (Coding Sequences) of Palaeoptera downloaded from NCBI in September 2022. Next, Maker v2.31.8 [51] was employed to predict the gene model for *A. torrens* using the gene training model from SNAP. For additional annotation, predicted protein sets were also tested using InterProScan v5.30-69.0 [52] and PSI-BLAST v2.4.0 [53] with EggNOG v4.5 [54].

Assembly quality was checked using several approaches. First, we calculated the 21-mer histogram using Jellyfish 1.1.12 [55], and the K-mer histogram was used to estimate genome size by genome size modeling using GenomeScope 1.0.0 [56]. Then, Illumina short reads were aligned against the assembly using the BWA-MEM algorithm. Depth of coverage was calculated using the DepthOfCoverage module of the Genome Analysis Toolkit (GATK) 4.2.0.0 [57], and Picard 2.18.11 [58] was employed to collect insert sizes. Finally, BUSCO 5.1.3 [59] was used to assess the completeness of assembly. This was performed with the AUGUSTUS program under the genome mode, using the insecta_odb10 lineage settings.

### 2.4. Analysis of Methylome Structure Based on Geographical Locations

To assess methylation variation across geographical sites, we performed a discriminant analysis of principal components (DAPC) [60]. This multivariate method, designed to analyze genetic structure from large genomic datasets, involves a two-step process. First, a principal component analysis (PCA) is conducted, which, in the present context, allowed summarizing variation in methylation profiles across individuals. Then, the PCA components describing variation between groups [61] are used in a discriminant analysis (DA). In this site-based approach, groups were defined according to sampling locations. The number of PCA components for the DA was set using K-1, as recommended by Thia [61] based on the principle that for K effective groups (here corresponding to the number of sites), only K-1 PCA axes capture the between-group variation. The significance of group effects on each discriminant axis was assessed using linear models, with significance adjusted by Benjamini–Hochberg correction. DAPCs were performed in R, using the ‘adegenet’ package. The Hellinger transformation was applied prior to the RDA analyses because Euclidean distances can misleadingly treat sites with no shared methylated markers as more similar to sites that have the same methylated markers but differ in their methylation levels (the Euclidean paradox) [62]. This transformation converts methylation levels into square-rooted relative values and effectively solves problems associated with the Euclidean paradox.

To identify the markers contributing most to the methylome structure, we calculated Mahalanobis distances for each marker along the retained discriminant axes. Markers showing significant distances using a false discovery rate (FDR) less than 5% were considered. We then performed iterative DAPC analyses using progressively larger subsets of these markers, ranked by decreasing Mahalanobis distances, and assessed the accuracy of group assignment. We determined the minimal set of markers required to achieve the highest level of accurate group assignment by performing segmented regressions [63] using the R ‘segmented’ package [64]. Models with one to 25 breakpoints were tested, and we selected the model based on the lowest AIC value. The first breakpoint, representing a marked change in assignment accuracy, was used to define the minimal informative marker set.

To further examine patterns of differentiation among populations, we conducted a hierarchical clustering analysis (UPGMA) based on population scores along the retained DAPC axes using the minimal informative marker set. The optimal number of clusters was determined by selecting the grouping scheme that maximized the correlation between the original distance matrix and binary matrices representing alternative partitions [65]. Finally, to characterize the methylation patterns of each site cluster, we identified hypermethylated markers characteristic of each group. For this, we used the indicator value IndVal [66], an index originally developed in community ecology to detect indicator species. IndVal combines both fidelity (frequency within a group) and specificity (exclusivity to the group). IndVal scores were computed for all markers in the minimal informative set using the R package ‘indicspecies’ (version 1.8.0). To avoid statistical circularity, no statistical test was applied. Instead, for each marker, we applied a Jenks natural breaks classification (with one breakpoint) to the distribution of IndVal values across groups. Markers with IndVal values above the breakpoint were considered characteristic of the group, indicating higher methylation levels in that group compared to the others.

### 2.5. Analysis of Main Discriminant Methylome Structure Patterns

To investigate dominant methylome structure, we conducted a second DAPC analysis. Groups were inferred de novo from the methylation data to identify the main discriminant methylation patterns across individuals, regardless of their geographical origin. The de novo groups were defined both with K-means clustering and by inspecting PCA scree plots concurrently. K-means clustering allocates observations into K clusters by minimizing the within-group sum of squares, with the optimal number of clusters determined by identifying the configuration with the lowest Bayesian Information Criterion (BIC) score [60]. PCA scree plot inspection was utilized to identify the elbow point, expected to occur when group structure is present in the data [61]. For the DAPC analyses, the number of PCA components selected for the DA step was set to K-1. In this case, the number of discriminant components was established by examining the DAPC scree plot and applying the elbow criterion. No inferential tests were conducted to assess differences among groups, as this would introduce circularity issues [61]. For each cluster thus identified, we then searched for representative markers by calculating IndVal scores for each marker across groups.

### 2.6. Functional Enrichment Analysis

To investigate the functional roles of genes associated with methylation markers, we first mapped the methylRAD markers that contributed most to methylome structure onto the *A. torrens* draft genome sequence with BLASTN Package: blast 2.5.0 (end-to-end option). Only loci with a single genomic hit were retained. Loci located within a gene body were assigned to that gene, while intergenic loci were associated with the closest gene located within 5 kb upstream or downstream. To characterize potential functional differences among groups, functional enrichment analyses were carried out separately for each set of hypermethylated markers identified for each population cluster and each de novo group. Functional enrichment analyses were performed using the Gene Ontology (GO) platform (geneontology.org), with *Drosophila melanogaster* as the reference database. When a gene name from our annotation was not recognized, we identified the putative *D. melanogaster* ortholog using OrthoDB v11 [67] and used its corresponding name and identification number. Fisher’s exact test was applied to identify biological processes overrepresented among genes linked to the hypermethylated markers of each group. GO terms with a false discovery rate (FDR) < 0.05 were first retained. As an additional filter, we only considered as significantly enriched those biological processes represented by at least three genes.

## 3. Results

### 3.1. Methylome Patterns Across Geographical Locations

The DAPC identified 11 significant DAs across populations. The overall assignment accuracy based on all markers reached 60%, indicating the presence of methylation structure across sites. We identified 3265 markers with significant Mahalanobis distances across the 11 DAs. From these, 291 formed the minimal dataset required to achieve the highest level of accurate assignment (Appendix A). Sequencing depth and methylation levels for these markers are reported in Appendix A. CCWGG markers again contributed disproportionately to this structure, representing >70% of the informative subset. With the reduced dataset of 291 informative markers, the overall assignment accuracy of the DAPC reached 50.5%. Hierarchical clustering of site scores revealed five clusters (Figure 2a). Two sites were clearly isolated: Site 7 (Group A, Figure 2a), which harbored the largest number of hypermethylated markers among all groups (i.e., 116, Figure 3a), and Site 9 (Group B, Figure 2a) with 58 hypermethylated markers (Figure 3a). Groups C and E each comprised five geographically dispersed sites (Figure 2a,b) and were characterized by 40 and 80 hypermethylated markers, respectively (Figure 3a). Group D, with 88 hypermethylated markers (Figure 3a), encompassed 60% of the sites (Figure 2) and was also broadly distributed across the study area (Figure 2b).

### 3.2. Main Methylome Structure Patterns

The PCA and K-means clustering identified six groups in the data (Appendix A). The second DAPC analysis, which used the six de novo methylation-based clusters as the grouping factor, confirmed strong differentiation in the methylome profiles among groups. The overall reassignment accuracy to the actual groups across the first two DAPCs reached 90%, exceeding 80% for all groups except for Group 4, which had a reassignment accuracy of 57%. A total of 2162 markers with significant Mahalanobis distances were identified. However, the first breakpoint in assignment accuracy occurred with 112 markers (Appendix A), which allowed an assignment accuracy of 87.36%. Although CCWGG sites only accounted for 8.61% of all filtered methylated sites, they represented over 70% of the most discriminating markers (i.e., 80 out of 112). Sequencing depth and methylation levels for these 112 markers are reported in Appendix A.

The de novo inferred groups showed no clear geographic separation but were instead broadly distributed across the study region (Figure 4a). Groups 1, 2, and 3 dominated, comprising nearly 75% of the individuals (i.e., 211 out of 285) and occurring in over 85% of the sampling sites (Figure 4a). Group 1 was the most frequent and widespread, being dominant in nearly half of the sites (i.e., 14), while Groups 2 and 3 each dominated in eight sites. Overall, Groups 1–3 displayed a moderate number of hypermethylated markers, ranging from 21 in Group 1 to 49 in Group 3 (Figure 3b, Appendix A). In contrast, Groups 4–6 were less common, being present in only 50 to 60% of the sites (Figure 4a), and none was dominant in any site. In groups 5 and 6, the number of hypermethylated markers (58 and 69, respectively) accounted for more than half of all discriminant markers (Figure 2b).

### 3.3. Genome Sequencing and Enrichment Analysis

For the genome sequencing of *A. torrens*, the PacBio sequencing produced 8,441,164 filtered reads with an average length of 9682 bp and an N50 of 13,293 bp, totaling 81.7 Gb of filtered bases. Complementary Illumina sequencing yielded 903,933,076 filtered reads of high quality (Q30 = 96.05%, Q20 = 99.12%) and an average GC content of 39.71%, corresponding to 136.4 Gb of filtered bases. The hybrid assembly and subsequent polishing resulted in a genome of 167.71 Mb distributed across 844 contigs, with an N50 of 3,017,901 bp and an average sequencing depth of 296.7× per base. Short-read mapping was used to assess the accuracy of the assembly, reaching a mean coverage depth of 623.57×. Of the 903,933,076 Illumina reads, 97.4% (880,469,513) successfully mapped to the final assembly. From the predicted transcriptome, a total of 17,342 genes were identified. Quality control analyses confirmed the robustness of the assembly. The average K-mer coverage for heterozygous bases was 34.36, and the estimated genome size (166.94 Mb) closely matched the assembled genome. BUSCO analysis indicated high completeness, with 97.01% of the expected orthologous genes detected.

Of the 291 discriminant markers identified in the geographical population-based DAPC analysis, 203 mapped to a single genomic location and were each associated with only one gene. Most were located within annotated genes and mRNAs, with only 23% found in intergenic regions (Appendix A). Among the gene-associated markers, most were located in exons, while only nine mapped to introns. At the population level, the most recurrent biological processes across the five groups (A–E) involved cellular regulation, morphogenesis, neurogenesis, metabolism, and gene expression control, highlighting a conserved set of core cellular and developmental mechanisms (Figure 2c, Appendix A for the list of all the significant GO terms). In contrast, processes associated with immune response, stress, and environmental stimuli appeared more selectively, confined mainly to groups D and E. The analysis also revealed strong contrasts in functional breadth, ranging from the narrow, specialized profiles of Groups B and C to the broad multifunctional profiles of Groups D and E, with Group A occupying an intermediate position characterized by general regulatory functions. Specifically, Group A was dominated by transcriptional and post-transcriptional regulation processes, reflecting basal gene-expression control. Group B emphasized metabolic and signaling regulation, modulating cellular communication and energy balance. Group C was characterized by neural and differentiation processes linked to tissue and structural specialization. Group D exhibited the broadest functional spectrum, integrating developmental, metabolic, immune, and potential adaptive responses within a multisystem regulatory framework. Finally, Group E integrated developmental and metabolic functions with stress and nutrient response pathways, which reflects coordinated regulation of growth and physiological adaptation. Cuticle development processes were also restricted to Groups D and E, further underscoring their shared involvement in external structural and adaptive functions. Another interesting feature of populations from groups C, D and E, is that they displayed genes involved in the regulation of biological quality, suggesting the influence of environmental conditions on the development of fitness-related traits. These results show a functional continuum from constitutive cellular regulation (A) to integrated developmental and adaptive control (E). Groups D and E had the most functionally diverse and interconnected profiles.

Among the 112 discriminant markers identified in the de novo DAPC grouping analysis, 86 could be mapped to a single genomic location, and each of these was associated with only one gene. Similar to the geographical population groups, most of these markers were located within genes (Appendix A), with the vast majority of them (90%) falling within exonic regions. Regarding the biological processes associated with these genes, the most recurrent processes across the six de novo methylation groups were cellular and metabolic processes, gene regulation, and protein localization, indicating that core regulatory and metabolic functions are widely shared. Among these, Groups 2, 5, and 6 exhibited the broadest functional diversity, while Groups 1 and 4 showed the most restricted profiles (Figure 4b, Appendix A for the list of all the significant GO terms).

Group 1 was characterized by moderate activity focused on cellular regulation, signaling, and metabolism, reflecting core regulatory and maintenance roles. Group 2 presented the widest range of biological processes, combining cellular, developmental, and regulatory functions, and acting as a developmental hub for differentiation-related pathways. Group 3 displayed an intermediate regulatory profile dominated by transcriptional and metabolic functions, with limited developmental and signaling categories but possible links to memory and sensory functions, suggesting more specialized regulatory roles. In contrast, Group 4 exhibited minimal functional diversity, restricted mainly to metabolic and signaling processes, consistent with a narrow metabolic focus. Group 5 stood out for its broad metabolic and regulatory activity coupled with a distinctive enrichment in epigenetic regulation and response to external stimuli, highlighting adaptive or stress-response regulation. Finally, Group 6 integrated developmental and regulatory processes, linking morphogenesis, signaling, and behavioral functions, and reflecting coordinated control of differentiation, possibly linked to late differentiation and neural functions. Overall, these results show a range from core regulatory and metabolic (Groups 1 and 4) to specialized transcriptional and neuronal activity (Group 3) and to highly integrative developmental Groups (2 and 6). Group 1 was characterized by moderate activity focused on cellular regulation, signaling, and metabolism, reflecting core regulatory and maintenance roles. Group 2 presented the widest range of biological processes, combining cellular, developmental, and regulatory functions, and acting as a developmental hub for differentiation-related pathways. Group 3 displayed an intermediate regulatory profile dominated by transcriptional and metabolic functions, with limited developmental and signaling categories but possible links to memory and sensory functions, suggesting more specialized regulatory roles. In contrast, Group 4 exhibited minimal functional diversity, restricted mainly to metabolic and signaling processes, consistent with a narrow metabolic focus. Group 5 stood out for its broad metabolic and regulatory activity coupled with a distinctive enrichment in epigenetic regulation and response to external stimuli, highlighting potential adaptive or stress-response regulation. Finally, Group 6 integrated developmental and regulatory processes, linking morphogenesis, signaling, and behavioral functions, and reflecting coordinated control of differentiation, possibly linked to late differentiation and neural functions. Overall, these results show a range from core regulatory and metabolic (Groups 1 and 4) to specialized transcriptional and neuronal activity (Group 3) and to highly integrative developmental Groups (2 and 6). Group 5 stands out because it had an adaptive profile that distinguishes it from the rest.

## 4. Discussion

In recent years, epigenomic studies in natural populations have progressed rapidly, revealing substantial levels of methylation variation across taxa [9,68,69]. Despite the increasing evidence, our understanding of the evolutionary importance of such variation in natural populations is still limited [8]. Recent studies indicate that genetic factors can be primary drivers of epigenetic divergence [8,18]. However, a substantial fraction of epigenetic variation arises independently of genetic differentiation and is better explained by environmental factors [8]. Such environmentally driven epigenetic variation could in fact be particularly common in natural systems [70,71,72]. Although our study did not directly evaluate environmental influences on *A. torrens* methylation patterns, several lines of evidence suggest that they might play an important role. First, our results reveal epigenetic differentiation among sites, while neutral *A. torrens* genetic differentiation in this system is extremely low (mean pairwise *F_ST_* of 0.0039 [33]), making a genetically driven explanation unlikely. Likewise, genetic structure cannot account for the main structuring methylation patterns we found, since the de novo groups did not show spatial arrangement. Finally, we found that CCWGG loci play a predominant role in the detected methylation patterns, and prior research indicates that non-CpG methylation contexts (such as CHG and CHH) tend to be more responsive to environmental factors than CpG methylation [16,73,74,75]. The large contribution of CCWGG markers to the observed methylation divergences therefore also points toward environmentally driven effects on *A. torrens* populations within the Limarí watershed.

The detection of significant geographical structure emphasizes the role of local environmental effects in shaping *A. torrens* methylation patterns. However, these pressures could not be linked to broad geographic trends or to expectations under a river continuum model [76], as population clusters did not follow a clear spatial arrangement along the watershed. Instead, several of the detected clusters encompassed sites scattered across the study area. The functional analysis of these population clusters supports that *A. torrens* populations experience widespread yet heterogeneous environmental stress across the watershed. Little has been described about water conditions along the Limarí watershed, but there is evidence that it varies spatially [33], and that pesticide contamination is extended, sometimes at high concentrations locally and with mixtures or compounds that differ among sites [29], which corroborates environmental heterogeneity. Although more notable for groups D and E, which comprise 23 of the 30 study sites, nearly all clusters were enriched in pathways potentially related to stress responses, including core regulatory processes for groups A and B and nervous system development for group C. Group D, which encompassed most sampling sites and thus reflects common water conditions across the watershed, was enriched in pathways related to cell signaling and communication, immune response, metabolism, neurogenesis, and responses to external stimuli and stress. This broad functional profile, integrating developmental, metabolic, immune, and adaptive responses, points to physiological adjustments to persistent environmental stressors, consistent with patterns observed in insects, including mayflies [77,78]. Similarly, group E, another geographically dispersed cluster, exhibited enrichment in metabolism, morphogenesis, neurogenesis, signaling, and physiological responses to stress, and nutrient variation. This combination of metabolic, developmental, and adaptive functions suggests ongoing adaptation to extrinsic pressures [78,79,80]. By their wide distribution along the watershed, these two groups support the view that environmental stress is a pervasive feature of the Limarí system. An interesting feature of groups D and E is that they are the only groups showing processes involved in cuticle development. Cuticle-related processes were previously suspected to contribute to pesticide resistance in *A. torrens* from the Limarí watershed by Gouin et al. [33], with the detection of two single nucleotide polymorphisms (SNPs) located within two genes implicated in cuticle-related processes and associated with pesticide concentrations. Our findings therefore suggest that pesticide contamination may have indeed affected these populations and provide more evidence towards the potential role of cuticle-related pesticide resistance/susceptibility mechanisms in this species. Additionally, our results revealed two site-based groups that were each represented by a single site (Sites 7 and 9), indicating unusually differentiated methylation profiles relative to all other populations. Previous evidence revealed pesticide contamination at both locations, which displayed the highest proportions of cultivated areas in the surrounding area (Site 7: 24%; Site 9: 32%). Compared with the other study sites (see Appendix 3 in [33]), Sites 7 and 9 also likely experience challenging environmental conditions, particularly during summer, with relatively high water temperatures (Site 7: 17.1 °C; Site 9: 17.9 °C) and low levels of dissolved oxygen (Site 7: 46.5%; Site 9: 61.2%). Previous monitoring revealed the presence of permethrin, a highly toxic pyrethroid insecticide, exclusively at Site 7 [33]. Such contamination could therefore act as a strong selective pressure, contributing to the extensive methylation response observed in *A. torrens* at this location. Indeed, Site 7 (Group A) exhibited the highest number of hypermethylated markers of all groups, which could indicate stress exposure driving extensive biological responses [77]. Its functional profile emphasizes the maintenance of essential biological functions, including transcriptional and post-transcriptional control, RNA/DNA metabolism, epigenetic regulation, and cellular organization, suggesting a generalized stress response aimed at preserving core cellular processes and integrity [81,82]. Stress-related genes included Sirt4, linked to metabolic homeostasis and fasting responses [83], and Trx, involved in antioxidant defense and cellular stress regulation [84]. At Site 9, Gouin et al. [33] reported traces below detection limits of deisopropylatrazine, an atrazine metabolite. More recently, the presence of two potent insecticides, carbofuran and tefluthrin, was detected in sediments at this site, at concentrations representing high risk for fish and invertebrates [29]. These two independent pieces of evidence of pesticide contamination strongly suggest that Site 9 is regularly subject to contamination by agrichemicals. Group B (Site 9), although showing a lower number of hypermethylated markers than Site 7, also displayed a distinct methylation profile suggestive of unique local pressures. Its functional signature reflects a metabolic–signaling coordination network aimed at maintaining physiological balance, including the regulation of metabolic activity, cell communication, apoptosis, and intracellular signaling. Such a profile indicates activation of compensatory and homeostatic mechanisms, consistent with chronic exposure to toxic stressors [79,85]. Finally, Group C presented a distinct functional signature centered on neurogenesis, morphogenesis, cellular differentiation, and tissue organization, consistent with a neural and differentiation-focused system supporting structural and functional specialization. The strong enrichment of this group in processes involved in the development of the nervous system is notable and can traduce the impact of contamination by insecticides, which can be neurotoxic and affect neuron development [86,87]. Pesticide contamination has been reported in four of the five sites composing this group, with carbofuran and tefluthrin insecticides in two of them [33]. More recently, the presence of acetochlor, bifenthrin, and tefluthrin was documented slightly upstream of Site 17, and malathion, carbofuran, lambda-cyhalothrin, and tefluthrin just upstream of Site 29, in concentrations that pose high risk to fish and macroinvertebrates [29], confirming the prevalence of highly toxic compounds with recognized neurotoxic effects like pyrethroids [87]. In addition, along with groups D and E, populations from group C displayed genes involved in the regulation of biological quality, which strongly suggests a broad influence of environmental conditions on the development of fitness-related traits. Temperature, which is an important stress factor in rivers of semi-arid region accentuated by water management for agricultural activities [88], can produce stress and affect energy use and development in insects, with recognized effects on body size and other fitness traits and consequences on population dynamics and evolutionary trajectories [89,90,91], including in freshwater species [92]. Because the samples for this study were taken in 2015, at the end of a five-year megadrought period in north-central Chile [33], our results clearly reflect the extreme environmental conditions in which most of the populations of this highly sensitive species were living at this time.

While our study contributes to the growing evidence that methylation patterns in natural populations often display geographical structuring [93,94,95,96,97,98], we found that the dominant methylation structure in *A. torrens* was not geographically constrained. We identified six highly differentiated methylation groups that co-occurred within sites, underscoring the value of exploratory approaches for revealing important dimensions of epigenetic variation that may be missed by analyses based on a priori population grouping. The coexistence of several methylation groups within the same site indicates that individuals inhabiting broadly similar environments can exhibit markedly different methylation profiles. Although little is known about *A. torrens* dispersal, a previous population genomic study suggested high gene flow among these populations [33]. In this context, downstream drift of aquatic nymphs, which is common in lotic ecosystems, could contribute to local admixture of methylation signatures from different locations with contrasting conditions. Considering that DNA methylation changes can also be transgenerational [99], adult dispersal between sites may further contribute to this pattern. Such divergence could also result from individual differences in biological status. It is well established that sex and developmental stages are important determinants of methylation patterns [100,101,102,103]. In support of this, Groups 2 and 6 displayed broad enrichment in regulatory and morphogenetic processes, which could correspond to specific developmental phases or physiological conditions characterized by distinct methylation signatures [103,104]. This may also be the case for Groups 1 and 3. Their functional enrichment in core cellular regulation (Group 1) and transcriptional and metabolic coordination (Group 3) is consistent with essential housekeeping functions rather than condition-specific responses [42,79,105]. Given the stage-specific developmental programs reported in Baetidae mayflies [106] and the fact that all *A. torrens* individuals were collected at the nymph stage within a relatively narrow temporal window, major ontogenetic differences are unlikely. However, because no phenotypic or fine-scale staging data were recorded, we cannot rule out the possibility that subtle developmental variation may partly underlie developmental-related methylation patterns. The predominance of Groups 1, 2, and 3 suggests that they correspond to widespread development stages, with Group 1 prevailing across most sites. In contrast, the low frequency of Group 6 rather suggests a transient or rare state. Fine-scale environmental heterogeneity, biotic and abiotic, is another possible driver of methylation divergence within sites. Depending on their position within the riverbed, *A. torrens* individuals can occupy a range of microhabitats differing in physical, chemical and species composition characteristics. For example, predation can affect genome-wide methylation patterns in freshwater invertebrates [71]. Alternatively, interactions between environmental pressures and intrinsic factors could modulate methylation patterns, with environmental effects depending on the timing of exposure during development or on sex-specific physiology [100,101,102]. Group 5 displayed functional enrichment in metabolic and stress-response pathways, suggesting environmentally induced methylation changes [102,107,108]. Given that this group occurred in roughly half of the sites but was represented by few individuals and was never dominant, it may reflect a condition triggered by relatively common environmental factors but expressed only in a rare developmental or physiological state. Together, these observations suggest that the methylation groups we identified likely represent both conservative, housekeeping-related regulation and stress-induced plasticity. This distinction also raises the possibility that some of the environmentally responsive methylation patterns we observed may align with hormetic processes. Hormesis involves adaptive responses triggered by mild exposure to a normally harmful stressor, leading to improved cellular or physiological function, and epigenetic regulation has been proposed as a central mediator of such responses [109]. The enrichment of signaling, metabolic, immune, and neurogenesis-related pathways in several groups, particularly D, E, and parts of B, is consistent with this form of stress-related plasticity. Although our data do not allow us to demonstrate hormesis directly, the coexistence within sites of methylation profiles associated with core cellular maintenance (e.g., Groups 1 and 3) alongside others enriched for induced stress responses suggests that *A. torrens* populations may rely on a spectrum of epigenetically mediated reactions to persistent environmental pressures. The possibility that repeated or chronic low-level exposure to contaminants and thermal stress in the Limarí watershed induces hormetic-like responses is especially relevant given the region’s documented history of recurrent pesticide inputs and extreme climatic conditions [29,33,41].

## 5. Conclusions

Our study provides new insights into the epigenetic landscape of natural populations. Although the dominant methylation variation was not spatially structured, site-level differences in methylation patterns still occurred in *A. torrens*, despite limited neutral genetic divergence. These results are consistent with a role for epigenetic mechanisms to local responses. The broad functional spectrum within population groups, primarily involving stress-related and regulatory processes, indicates pervasive and heterogeneous environmental pressures within the Limarí watershed in Chile, emphasizing the vulnerability of arid and semi-arid watersheds to ongoing climatic change and persistent human disturbances. Together, our results emphasize the suitability of mayflies as model organisms for studying environmental pressures in freshwaters. However, the presence of individuals with divergent methylation profiles within sites also demonstrates that confounding factors, which could be linked to dispersal, intrinsic biological processes and/or fine-scale environmental heterogeneity, may obscure environmentally driven signals in wild populations. Valuable insights into the environmental drivers and evolutionary relevance of methylation patterns would be gained from complementary laboratory and intergenerational experiments designed to isolate the effects of specific environmental stressors.

## Figures and Tables

**Figure 1 biology-15-00090-f001:**
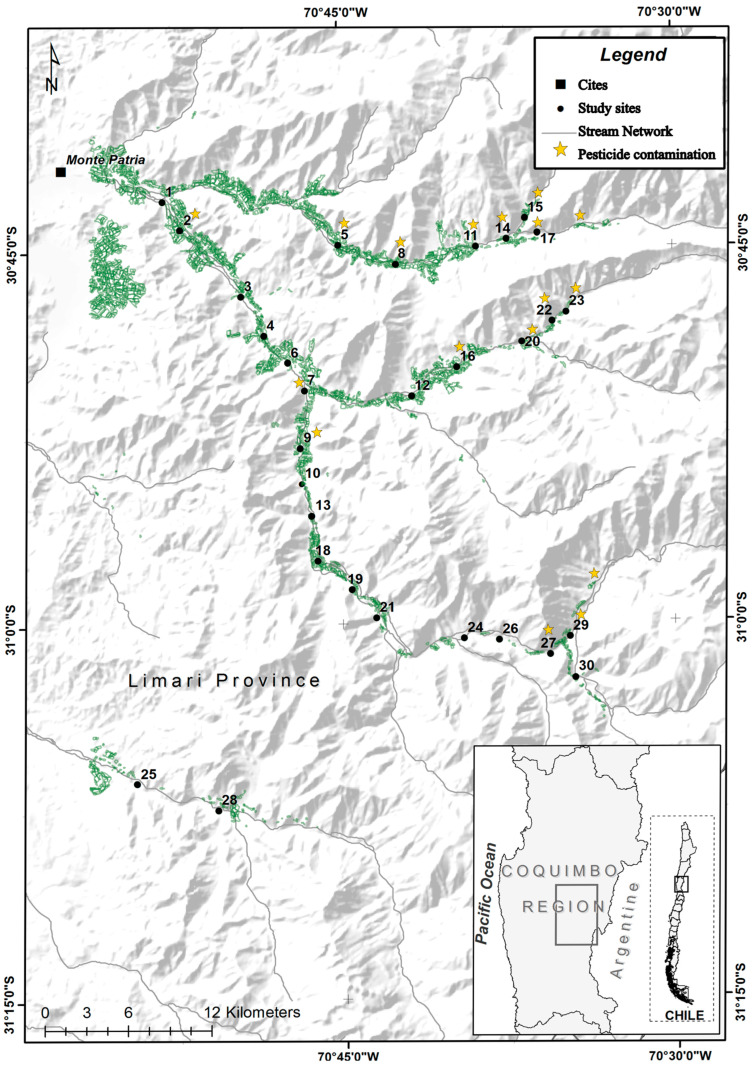
Map of the study area, with the 30 sampling sites of the mayfly *Andesiops torrens* along the Limarí watershed. Green areas represent agricultural land cover, and yellow stars indicate sites where pesticide contamination was detected in previous studies [29,33].

**Figure 2 biology-15-00090-f002:**
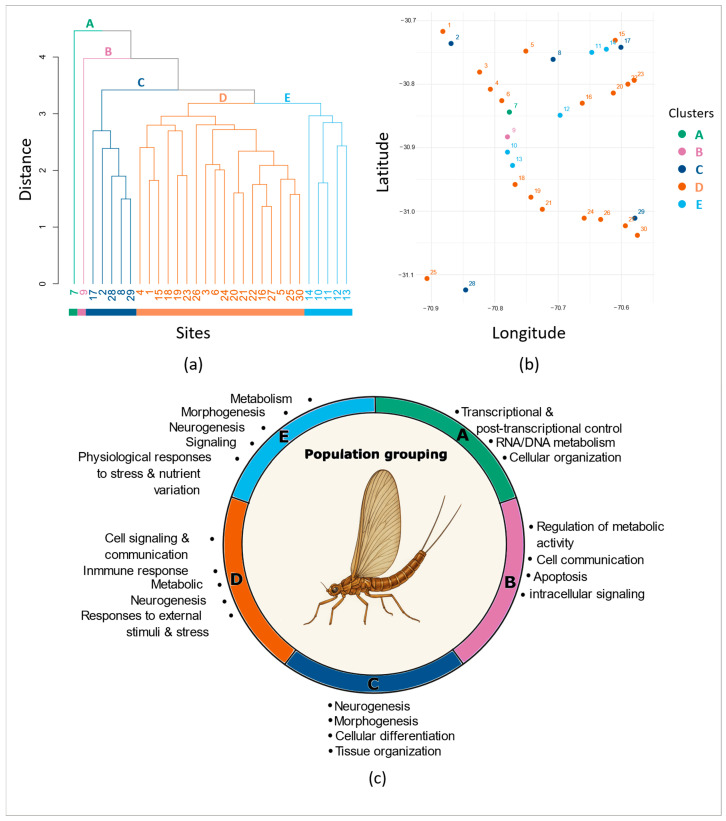
Population clustering from (**a**) DAPC site scores, (**b**) corresponding geographical locations (Information on study locations and ID number correspondence are given in Appendix A) and (**c**) functional profiles of population-based methylation groups (A–E), where each colored arc represents the dominant biological processes enriched in each population group, as identified by GO enrichment analyses.

**Figure 3 biology-15-00090-f003:**
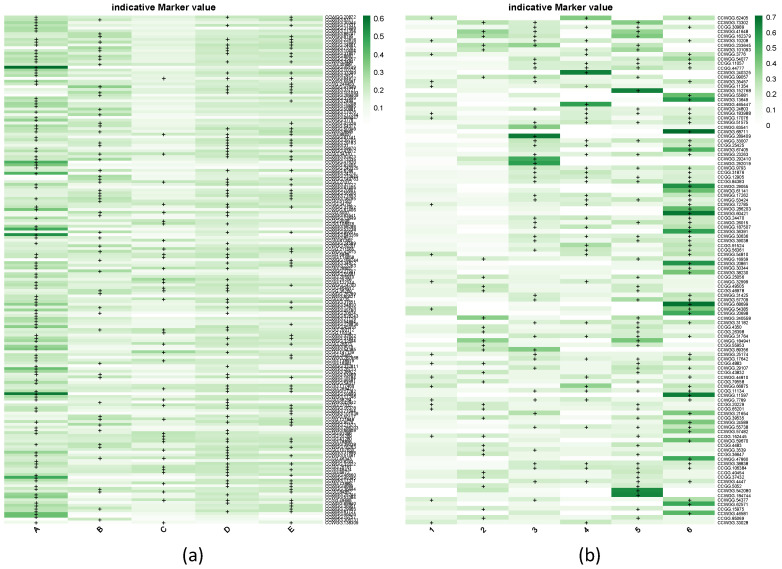
IndVal score heatmaps for discriminant methylation markers in (**a**) the five geographic population clusters (A–E) and (**b**) the six de novo DAPC groups (1–6). Hypermethylated markers defined by Jenks natural breaks are marked with “+”.

**Figure 4 biology-15-00090-f004:**
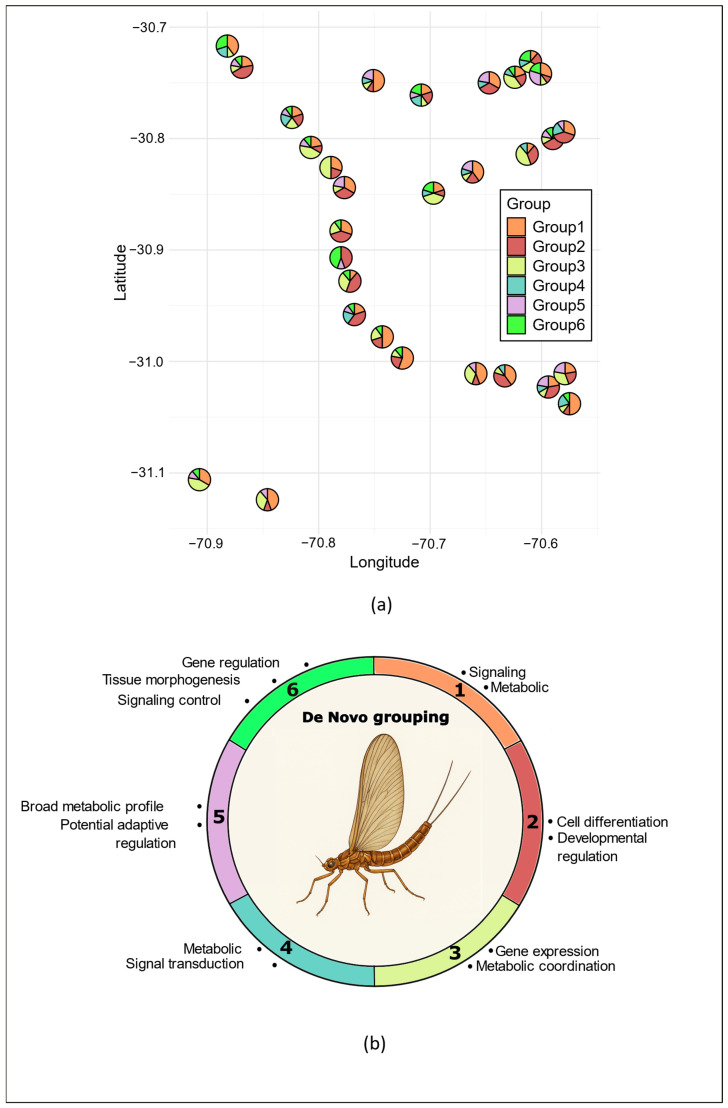
Geographic distribution of de novo groups across sampling sites of *Andesiops torrens*. (**a**); the pie charts at each site represent the proportion of individuals assigned to each of the six de novo groups. (Study locations are identical to Figure 1, and information is given in Appendix A), and (**b**) functional profiles of de novo methylation groups (1–6), where each colored arc represents the dominant biological processes enriched in the six de novo methylation groups, as identified by GO enrichment analyses.

## Data Availability

MethylRad data with structuring effects used in the study are provided as Appendix A. All sequencing data generated in this study, including the raw sequencing reads (Illumina: SRR36185091; PacBio: SRR36185090) and the genome assembly are available from NCBI under Bioproject accession [PRJNA1368467]. A version of the genome assembly and annotation data (genome with rich annotation in various formats) is also available on Zenodo (10.5281/zenodo.17702558).

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
