# Peer review of "Patterns and Functional Insights of DNA Methylation Variation in a South American Mayfly Across an Agriculturally Impacted Semi-Arid Watershed"

_biology, 2025, doi:10.3390/biology15010090_

Round 1

Reviewer 1 Report

Comments and Suggestions for Authors

Review of  “Epigenetic Insights Into Environmental Pressures in Natural Populations of a South American Mayfly“

The research into methylation profiles in wild populations is interesting and important, especially in non model species, where patterns, functions and impact from environment and development etc. is not known. Studying non-nucleotide variation is especially interesting in organisms where genetic differentiation is low, since the assumption is, that responses to environmental pressures must come from something else.

However, in my view, relating methylation patterns to broad biological functions of genes becomes really interesting if the environmental connection is there. In this paper, it seems that differing methylation profiles may underlie differing functions and responses based on GO-terms, but the connection with the environment is lacking (or anecdotal). It is non the less interesting that, under the assumption that there is a marked effect of methylation on gene function, that there seem to be differing responses to the environment across the watershed.

Thus, my main worry is that one of the main take aways (and present in title) is that methylation patterns give insights into the environmental pressures in a watershed, and this is not fully backed up. The connection between methylation and environmental pressure stems from another paper, while this one mainly describes the methylation patterns across the watershed, and tries to link it with functions of the related genes. The direct connection with the environmental pressure is lacking, or at best unclear. I suggest measuring and including the pressures, or reframe the story to focus on the seemingly differering methylation profiles, and that there seem to be a heterogenous shift in the functions of genes enriched in methylations across the watershed. Also, the impact of lifestage on methylation pattern does not get the attention that it might warrant.

Other than this, there are places where it is unclear what it meant, and other places where the logic flow is not robust, see below comments.

See my detailed comments below:

Line 56-57. I think references to the claim is lacking.

86-88. To my knowledge, it is not clear that there is a targeted effect of stress related methylation, and therefore the link to adaptive potential is unclear.

107-109. The fact there is low genetic diversity does not suggest a role for methylation. It suggest that there are other mechanisms, among those methylation may be one such mechanism. I suggest to reformulate.

113-114. True, the function remains unclear, so take care not to conclude too strongly based on this.

120-124. I am not sure it illustrates that methylation can uncover environmental preesures, providing that you do not have the pressures measured. At best it highlights the possibility.

135-137. Are these papers based on the same populations? Then you do have measures of pesticides and water related environmental data. Why not use it?

155-156. But you do have one now? Will that improve the analysis, if rerun?

251-252. I believe it shows the most discriminating patterns? Or the patterns maximizing group difference, not the most prevalent one?

280-281. It seems somewhat arbitrary using three as a threshold for enriched or not. In other contexts, this is done by taking a larger set of genes, and compare the subset, to see if the subset is more enriched in certain categories. It is not clear if this was done, and this “3 genes” is an additional filter? If so, please state both the subset and the larger set of genes.

314-316. What is the genome wide distribution? Were some methylation calls done in lowly methylated areas, where calls are less secure in methylRad?

  1. Do you mean separation?

Figure 3. An idea: could a construct analysis be employed to inspect "methylation layers" in the samples ? it will also group the patterns, and indiate which pops share how much of each of these patterns. It will give you a dynamic picture of shared methylation layers, (something defining a specific region)? As methylation is more dynamic than the SNP pop frequencies the method was developed for, this should of course be reflected in interpretation, such that it does not show barriers in the env, but more like clusters of differing methylation layers. It may yield another layer of information, since there is a focus on sharedness, less so on discrimination?

Also, figures should be color-corrdinated. So fig 3 and fig 5 should show same colors for same cluster? Same for 1 and 4.

  1. What is meant by dominant here? Enriched?
  2. What is the adaptive profile? I would be carefull in including adaptation.

420-422. Half the references are concerning animals here (though definitely relevant for the paper), but the statement in concerning plants. Consider revising or replacing.

421-422. Why is this unlikely? I know the I comes in later in the paragraph, but it disrupts flow

423-424. You find geographically unstructured pattern, and say it is contrary to a pattern driven by genetic differences. But genetic differences in this system is low? Explain what you mean.

424-426. This should probably be more clearly explained.  Tre methylation pattern was most likely not composed of several individuals, but the pattern was found in individuals from distant locations?

427-429. This should probably be in connection with the "unlikely in our system" above? This feels like a repeat, but with the explanation.

430-433. Maybe it is just a formulation issue, but the first sentence here states that you find effects driven by environment. Yet, you find no structure according to geography, and you do not present data from the environment, except stating it is heterogenous.

So the claim comes from the fact that the primary driver of differentiation is CCWGG context, that from other studies are found to be enviromentally responsive. This seems circular. Your finding does not back up the environmental responsivenes results from other studies, in my view you rely on the studies to conclude your findings.

I believe it should be reformulated so there is no confusion about line of evidence.

  1. Sometimes you write geographically unstructured and sometimes geographically structured. Are you referring to something else than methylation pattern here? It is unclear what you refer to here as geographically structured.

443-445. Indeed this study would be stronger with an environmental link.

  1. Why does the study suggest that?

472-477. Could you look more specifically in stress response genes (or other interesting subsets) directly? If this is your claim, maybe these genes specifically should be differentially affected by methylation across these sites?

  1. Environmental information like this is very interesting and would have been much stronger if quantified and visualized across study sites, instead of anecdotal, or referring to a supplement of another paper.

530-532. This is one reason I believe there should not be put too much emphasis on Go terms, since they can envelope so many differing cellular processes. However, for showing there is a change in the the general signal is fine.

  1. This one I agree with.
  2. Not quite agree on that there is a stucture? Maybe unstructured methylation pattern? Or be more clear what you mean.

Lastly, supplementary figures are very difficult to read. Please provide in better resolution.

Author Response

Reviewer 1:
Review of  “Epigenetic Insights Into Environmental Pressures in Natural Populations of a South American Mayfly“

The research into methylation profiles in wild populations is interesting and important, especially in non model species, where patterns, functions and impact from environment and development etc. is not known. Studying non-nucleotide variation is especially interesting in organisms where genetic differentiation is low, since the assumption is, that responses to environmental pressures must come from something else.

Comment #1

However, in my view, relating methylation patterns to broad biological functions of genes becomes really interesting if the environmental connection is there. In this paper, it seems that differing methylation profiles may underlie differing functions and responses based on GO-terms, but the connection with the environment is lacking (or anecdotal). It is non the less interesting that, under the assumption that there is a marked effect of methylation on gene function, that there seem to be differing responses to the environment across the watershed.

Thus, my main worry is that one of the main take aways (and present in title) is that methylation patterns give insights into the environmental pressures in a watershed, and this is not fully backed up. The connection between methylation and environmental pressure stems from another paper, while this one mainly describes the methylation patterns across the watershed, and tries to link it with functions of the related genes. The direct connection with the environmental pressure is lacking, or at best unclear. I suggest measuring and including the pressures, or reframe the story to focus on the seemingly differering methylation profiles, and that there seem to be a heterogenous shift in the functions of genes enriched in methylations across the watershed.

We agree with the reviewer that, in its previous form, the manuscript placed too much emphasis on environmental pressures without providing direct environmental data analyses to support such links. Although pesticide and water-quality data exist for the same sampling sites (Gouin et al. 2019; 2025), these datasets are not sufficiently suited for a robust correlative approach. Measurements are sparse, heterogeneous across sites, and cover numerous variables for which only two temporal measurements were available, which limits their reliability for associating site-level conditions with highly variable methylation data. In addition, methylation patterns in A. torrens show substantial individual-level variability within sites. Environmental data could thus not be meaningfully paired with the de novo methylation groups. Given these limitations, we chose not to perform new correlation analyses, as they would risk being statistically weak or misleading. Instead, we revised the manuscript to accurately reflect what can be supported by the data. We modified the title, the abstract, the hypothesis, and several sections of the discussion to remove clear mechanistic claims about environmental pressures. Interpretations have been reframed to focus on describing spatial and de novo methylation structuring and on the functional differences among methylated loci, without inferring direct causal links with local environmental stressors.

We have clarified these points throughout the manuscript and hope that the revised framing now aligns with the scope and strength of the available data. We also downplayed the adaptive perspective, by removing “adaptive responses“ from keywords and “and may thus represent a pervasive adaptive mechanism” from the text.

Comment #2

Also, the impact of lifestage on methylation pattern does not get the attention that it might warrant.

The impact of lifestage is now better discussed We have added information in the discussion using the Gouin et al. (2019) paper revealing a connected metacommunity in this watershed, and discussed how downstream drift of aquatic nymphs and dispersal of adults can affect the patterns of DNA methylation (L.579-584). We also included a reference of Almudi et al. (2020) to specify that gene expression patterns are stage-specific in mayflies from the Baetidae family (L.593-594), and give more arguments that our sampling strategy is unlikely to have produced strong effects on the variation of our DNA methylation profiles.

Other than this, there are places where it is unclear what it meant, and other places where the logic flow is not robust, see below comments.

See my detailed comments below:

Comment #3

Line 56-57. I think references to the claim is lacking.

We added appropriate references to support the statement (L.61).

Comment #4

86-88. To my knowledge, it is not clear that there is a targeted effect of stress related methylation, and therefore the link to adaptive potential is unclear.

We understand the reviewer’s point and have therefore revised the sentence to avoid this implication and now focus on the value of methylation screening as an approach to track environmental pressures. (L. 90-91).

Comment #5

107-109. The fact there is low genetic diversity does not suggest a role for methylation. It suggest that there are other mechanisms, among those methylation may be one such mechanism. I suggest to reformulate.

We revised the sentence to clarify that mechanisms other than genetic variation may contribute to phenotypic differences. (L115-117)

Comment #6

113-114. True, the function remains unclear, so take care not to conclude too strongly based on this.

As explained in our response to Comment #1, we have revised the manuscript to maintain a cautious interpretation of our results and have downplayed conclusions regarding environmental impacts and biological relevance throughout the text.

Comment #7

120-124. I am not sure it illustrates that methylation can uncover environmental pressures, providing that you do not have the pressures measured. At best it highlights the possibility.

We understand the reviewer’s concern and agree. We reformulated the sentence to avoid overstatement L128-130)

Comment #8

135-137. Are these papers based on the same populations? Then you do have measures of pesticides and water related environmental data. Why not use it?

As explained in our response to comment #1, although the cited studies are based on the same populations and samples, the available environmental data are not adequate for a robust correlative analysis. In addition, such analyses would not be applicable to the de novo methylation clusters, which were a central focus of our study. For these reasons, we revised the manuscript to avoid strong claims about environmental causation and focused instead on describing methylation patterns and their functional context.

Comment #9

155-156. But you do have one now? Will that improve the analysis, if rerun?

The mention of the reference genome at this step was inappropriate since it is not use at this stage and has been removed (L. 169). MethylRAD data were generated as described in Wang et al. (2015). Because methylRAD fragments are very short, not all loci map to the genome, as reflected in our results (L337–338; L365–366). We now cite Wang et al. (2015) to clarify the correct methodological basis (L.179)

Comment #10

251-252. I believe it shows the most discriminating patterns? Or the patterns maximizing group difference, not the most prevalent one?

The reviewer is right. We revised the section title and corresponding sentence to reflect that the analysis identifies the main discriminant rather than prevalent patterns. (L. 277, 279, L. 336)

Comment #11

280-281. It seems somewhat arbitrary using three as a threshold for enriched or not. In other contexts, this is done by taking a larger set of genes, and compare the subset, to see if the subset is more enriched in certain categories. It is not clear if this was done, and this “3 genes” is an additional filter? If so, please state both the subset and the larger set of genes.

Our phrasing was not clear. The “3 GO terms” criterion for determining whether a biological process was enriched was in fact used as an additional filter. We first performed the enrichment analysis by comparing our GO terms with the reference gene set, as usually performed, which resulted in numerous significantly enriched processes. Then, as a complementary criterion, we decided to select only those significantly enriched biological processes that also contained a minimum of three GO terms associated with them. We have improved the text to avoid any confusion. (L.307-310).

Comment #12

314-316. What is the genome wide distribution? Were some methylation calls done in lowly methylated areas, where calls are less secure in methylRad?

The genome wide distribution of DNA methylation in this species is not the focus of the present study. Among the significant loci we detected and mapped, which are not many, there is no sign of methylRAD marker agglomeration in some specific regions. Sometimes two markers can map closely, within less than 70 kb, but no pattern is found along the genome, as observable from Tables S4 and S6.

Comment #13

  1. Do you mean separation?

Yes, we meant separation. We revised the sentence accordingly. (L349)

Comment #14

Figure 3. An idea: could a construct analysis be employed to inspect "methylation layers" in the samples ? it will also group the patterns, and indiate which pops share how much of each of these patterns. It will give you a dynamic picture of shared methylation layers, (something defining a specific region)? As methylation is more dynamic than the SNP pop frequencies the method was developed for, this should of course be reflected in interpretation, such that it does not show barriers in the env, but more like clusters of differing methylation layers. It may yield another layer of information, since there is a focus on sharedness, less so on discrimination?

We agree with the reviewer that it could be an interesting idea the objective of the present paper is to show that DNA methylation patterns can be structured within and between sampling sites, for which we provide robust evidence already. In addition, the timeframe to do it and incorporate the results in the manuscript was too short according to the deadline we were provided.

Comment #15

Also, figures should be color-corrdinated. So fig 3 and fig 5 should show same colors for same cluster? Same for 1 and 4.

We made the corresponding changes to ensure consistent color coordination across the figures.

Comment #16

  1. What is meant by dominant here? Enriched?

We removed “dominant” from the figure legend and only refer to “enriched”. Legend figure Figures 5 and 6.

Comment #17

  1. What is the adaptive profile? I would be carefull in including adaptation.

Regarding Group 5, the adaptive component of the profile is given by the enrichment in the functions epigenetic regulation of gene expression (3 genes) and regulation of response to external stimulus (7 genes), as shown on Table S5. These functions are found in a significant number of genes, so the adaptive potential is clearly found, and specific to this group. Nonetheless, we agree that this is not the principal specificity of the profile for Group 5, which shows mainly a broad metabolic profile. This bias was not strongly expressed in the main text, but was the main definition of the profile in Figure 5. We have slightly modified the text to indicate that this profile also highlights “potential” adaptive or stress-response regulation (L. 398, 457), and changed the main profile for this group in Figure 5. Figure 5 now says that Group 5 has a broad metabolic profile with potential adaptive regulation.

Comment #18

420-422. Half the references are concerning animals here (though definitely relevant for the paper), but the statement in concerning plants. Consider revising or replacing.

We revised and simplify the text of this section. The statement about plants was eliminated (L468-474).

Comment #19

421-422. Why is this unlikely? I know the I comes in later in the paragraph, but it disrupts flow

We revised the text to clarify why genetically driven methylation patterns are unlikely in our study system and improved the flow of the paragraph. Our revision also addresses comment 61 from Reviewer 3 (L472-484).

Comment #20

423-424. You find geographically unstructured pattern, and say it is contrary to a pattern driven by genetic differences. But genetic differences in this system is low? Explain what you mean.

We revised the whole paragraph to clarify our arguments of why genetically driven patterns are unlikely in this system (L472-484).

Comment #21

424-426. This should probably be more clearly explained.  Tre methylation pattern was most likely not composed of several individuals, but the pattern was found in individuals from distant locations?

We reformulated the corresponding section to clearly state that the de novo methylation groups showed no spatial structure (L.478).

Comment #22

427-429. This should probably be in connection with the "unlikely in our system" above? This feels like a repeat, but with the explanation.

We reorganized the paragraph so that this explanation appears first in our argumentation (L.474-477).

Comment #23

430-433. Maybe it is just a formulation issue, but the first sentence here states that you find effects driven by environment. Yet, you find no structure according to geography, and you do not present data from the environment, except stating it is heterogenous.

So the claim comes from the fact that the primary driver of differentiation is CCWGG context, that from other studies are found to be enviromentally responsive. This seems circular. Your finding does not back up the environmental responsivenes results from other studies, in my view you rely on the studies to conclude your findings.

I believe it should be reformulated so there is no confusion about line of evidence.

We agree with the reviewer and revised this section to avoid overstatement. The text now clearly indicates that our study did not directly evaluate environmental influences and that they are a plausible interpretation rather than a demonstrated cause. We agree also that our results do not provide further evidence that non-CpG markers are more responsive to environmental influences than CpG markers. We propose that, given previous evidence indicating that non-CpG markers tend to be more responsive to environmental influence, the predominance of CCWGG markers in the observed methylation patterns may point toward environmentally driven effects. We reformulated the section to clarify this point (l. 478-484).

Comment #24

  1. Sometimes you write geographically unstructured and sometimes geographically structured. Are you referring to something else than methylation pattern here? It is unclear what you refer to here as geographically structured.

We were referring as “geographically unstructured” to the main structuring patterns as discovered by the de novo groups and as “geographically structure” to the differences observed between populations. We reformulated the paragraph to avoid confusion (L. 478)

Comment #25

443-445. Indeed this study would be stronger with an environmental link.

We understand the reviewer comment. As explained in our responses to Comments #1 and others, available environmental data are too sparse and heterogeneous to support robust correlative analyses, and integrating them was beyond the scope of the present work. We have therefore revised the manuscript to ensure that our objectives, hypotheses, and interpretations are fully consistent with the data analyzed. These revisions clarify the limitations of environmental inference and avoid overstating environmental effects.

Comment #26

  1. Why does the study suggest that?

The explaination of the statement was provided in the following sentences but we understand that the reviewer was implying the our statement disrupted flow. Thus, we eliminated the sentence and reordered the paragraph to improve its flow (L.518-548).

Comment #27

472-477. Could you look more specifically in stress response genes (or other interesting subsets) directly? If this is your claim, maybe these genes specifically should be differentially affected by methylation across these sites?

Yes indeed, some of these genes can be involved specifically in responses to stress, such as Sirt4 (response to fasting, ensuring metabolic homeostasis and longevity) and Trx (antioxidant defense, regulation of stress). We have added this information (L. 535-536) and the relevant references.

Comment #28

  1. Environmental information like this is very interesting and would have been much stronger if quantified and visualized across study sites, instead of anecdotal, or referring to a supplement of another paper.

As explained in our response to Comment #1, available environmental data for these sites are too sparse and heterogeneous to allow reliable site-level quantification or modelling, which is why they were not incorporated in this study.

Comment #29

530-532. This is one reason I believe there should not be put too much emphasis on Go terms, since they can envelope so many differing cellular processes. However, for showing there is a change in the the general signal is fine.

GO term enrichment analyses are widely used in functional genomics to summarize the main functional tendencies within sets of genes. We agree that GO categories can group together diverse cellular processes, and therefore must be interpreted with caution. In our study, we used GO enrichment only to identify broad functional patterns across methylation-based groups, rather than to infer specific mechanistic pathways. We revised the text to better reflect this interpretative caution and to clarify that our conclusions focus on general functional trends rather than detailed process-level inferences.

Comment #30

  1. This one I agree with.

We retained the sentence as written and made adjustments thoughout the manuscript to highlight this point as one of the main objectives of the study.

Comment #31

  1. Not quite agree on that there is a stucture? Maybe unstructured methylation pattern? Or be more clear what you mean.

Although main methylation patterns are largely unstructured geographically (as demonstrated by the de novo analysis), the DAPC using sampling sites as grouping factors was significant, with an overall assignment accuracy of 60%, indicating some population-level structure. We have reformulated the sentence to clarify this point in the manuscript. (L.633-635)

Comment #32

Lastly, supplementary figures are very difficult to read. Please provide in better resolution.

 We upscaled Supplementary Figures S2 and S3 to ensure readability.

Reviewer 2 Report

Comments and Suggestions for Authors

The study by Bertin et al. studies methylation differences between wild populations of the mayfly Andesiops torrens in the Limari River watershed. They find five geographic location clusters based on methylation data which exhibited methylation differences located within genes associated with immune response, stress and environmental stimuli. They also find six clusters based purely on methylation patterns. Finally, they generate a genome for the organism in question.  Whilst the use of methylRAD is suboptimum, the number of samples used in this study warrants this reduced representation approach. Overall the paper is clearly written, well referenced, contains a clear methodology and finds novel results. For this enjoyable read, I outline a few questions below.

Specific Points:

  1. In lines 159-161 the authors state “DNA methylation levels were estimated from the normalized read depth (reads per million, RPM) at each site, calculated as RPM = (reads coverage per site/high quality reads per library) × 1,000,000”. What is the measure of DNA methylation levels? If fractional methylation, what is the purpose of normalizing read depth? Fractional methylation levels are already a percentage so wouldn’t need normalization, and doing so may artificially alter methylation levels. Perhaps there is an explanation I am missing, but in Supplementary table 2, sample HLJ_5 for CCGG.10024 has a sequencing depth of 1 but a methylation level of 0.18. Similar critiques for Supplementary table 3.

  1. Similarly for lines 221 and 222, could you specify the reasons for using Hellinger transformation?

  1. If my understanding of the coverage is right, no threshold has been applied to include sites in the analysis? Is it useful to interpret a loci’s methylation if it has a coverage of 1?

  1. It would be useful to have a bit more background on mayfly methylation, although this is probably difficult given the authors had to assemble a de novo genome. But is there any information on which Dnmt’s they posses?

  1. 292-293 states “Hierarchial clustering of site scores revealed 6 clusters (Figure 1a)”, but figure 1a and 1b show 5 clusters (A-E)

  1. Given there is little methylation data on this species to this point, it would be worth describing methylation location beyond just the identified markers (351-352). What are the main genomic locations of the 665,987 CCGG and 711,160 CCWGG sites?

  1. It would be interesting to hear if any matching phenotypic data exists for this dataset. For example, for de novo methylation cluster 2 there are developmental terms. Could these arise from earlier / later stage samples? Were any of the de novo clusters predominantly populated by samples of a particular date? I know this would be hard to collect in retrospect but I think it would be an interesting observation.

Minor points:

  • Line 23 – Given it’s a methylation paper I would avoid the use of imprint here.
  • In line 83 it would be useful to have references to the stated stressors
  • 156 – The author says there was no reference genome available at the time. Are there plans to release the genome generated in this paper?
  • 156 – I know it’s pedantic but please state high quality for first HC abbreviation
  • Line 307 – It would be worth reiterating to the reader this is a second PCA and describing the difference between the two, especially given the confusion between 5/6 clusters given that this has 6 groups.

The study provides interesting results, and with clarification on the points above would make a suitable fit for the journal.

Author Response

Reviewer 2:

Comment #33

The study by Bertin et al. studies methylation differences between wild populations of the mayfly Andesiops torrens in the Limari River watershed. They find five geographic location clusters based on methylation data which exhibited methylation differences located within genes associated with immune response, stress and environmental stimuli. They also find six clusters based purely on methylation patterns. Finally, they generate a genome for the organism in question.  Whilst the use of methylRAD is suboptimum, the number of samples used in this study warrants this reduced representation approach. Overall the paper is clearly written, well referenced, contains a clear methodology and finds novel results. For this enjoyable read, I outline a few questions below.

Specific Points:

Comment #34

In lines 159-161 the authors state “DNA methylation levels were estimated from the normalized read depth (reads per million, RPM) at each site, calculated as RPM = (reads coverage per site/high quality reads per library) × 1,000,000”. What is the measure of DNA methylation levels? If fractional methylation, what is the purpose of normalizing read depth? Fractional methylation levels are already a percentage so wouldn’t need normalization, and doing so may artificially alter methylation levels. Perhaps there is an explanation I am missing, but in Supplementary table 2, sample HLJ_5 for CCGG.10024 has a sequencing depth of 1 but a methylation level of 0.18. Similar critiques for Supplementary table 3.

Maybe our sentence was not clear. Relative DNA methylation levels were calculated using the RPM formula, which is the normalized read depth, as performed in Wang et al. (2025). This is how it is done with methylRAD data. So, our reported values are not percentages, but a relative quantification of methylRAD data based on sequence reads. We have improved the text in the manuscript to avoid any confusion (L.183-185).

Comment #35

Similarly for lines 221 and 222, could you specify the reasons for using Hellinger transformation?

Euclidean distances can misleadingly consider sites with no shared methylated markers as more similar than sites that share markers but differ in their methylation levels (the Euclidean paradox). The Hellinger transformation, which in this case converts methylation levels into square-rooted relative values, effectively resolves the distortions associated with this paradox. We have now added a justification for the use of the Hellinger transformation in the manuscript (L.245-250).

Comment #36

If my understanding of the coverage is right, no threshold has been applied to include sites in the analysis? Is it useful to interpret a loci’s methylation if it has a coverage of 1?

We have decided to apply no threshold in this study because we are comparing the relative DNA methylation levels between individuals. In this context, individuals with a low number of reads will display lower levels of DNA methylation than individuals with higher levels of reads. This is proportional, and should not bias any comparison. This is actually safer and more robust than trying to apply a coverage threshold to determine if a site is methylated or not (i.e. to obtain qualitative data). In our study, most of the discriminant loci display substantial levels of average coverage, that vary between individuals (see Tables S2 and S3 in which we have added some basic statistics). Only 7.6% of the loci harbor average coverage values below 10 in our population dataset (Table S2), and 3.6% for the de novo dataset (Table S3), yet they still display relevant coefficients of variation and are therefore informative. We hope we have clarified the reviewer’s concern.

Comment #37

It would be useful to have a bit more background on mayfly methylation, although this is probably difficult given the authors had to assemble a de novo genome. But is there any information on which Dnmt’s they posses?

Indeed, obtaining information about mayflies is challenging. To date, only four genomes have been published (Cloeon dipterum, Neocloeon triangulifer, Ephemera danica, and Siphlonurus alternatus), and the information available on the biology and ecology of our study species, A. torrens, is still limited. In A. torrens’ draft genome, only the enzyme Dnmt1, known as the “maintenance methyltransferase” (Provataris et al., 2018), was identified. This is consistent with the literature, which reports only the presence of Dnmt1 within the Ephemeroptera order (Bewick et al., 2017). However, DNMT1 may exhibit functional diversity since it can be activated under various conditions, and also act as a de novo DNA methyltransferase (Fatemi et al., 2002), which could be a compensatory mechanism for the evolutionary loss of DNMT3 in insects. Information has been included in the manuscript (L147-153)

Referencias:
- Provataris, P., Meusemann, K., Niehuis, O., Grath, S., & Misof, B. (2018). Signatures of DNA methylation across insects suggest reduced DNA methylation levels in Holometabola. Genome biology and evolution, 10(4), 1185-1197.
- Fatemi, M., Hermann, A., Gowher, H., & Jeltsch, A. (2002). Dnmt3a and Dnmt1 functionally cooperate during de novo methylation of DNA. European journal of biochemistry, 269(20), 4981-4984.
- Bewick, A. J., Vogel, K. J., Moore, A. J., & Schmitz, R. J. (2017). Evolution of DNA methylation across insects. Molecular Biology and Evolution, 34(3), 654–665. 
https://doi.org/10.1093/molbev/msw264

Comment #38

292-293 states “Hierarchial clustering of site scores revealed 6 clusters (Figure 1a)”, but figure 1a and 1b show 5 clusters (A-E)

The reviewer is correct: we mistakenly stated six clusters, whereas the results show five. We have updated the text accordingly. (L.322)

Comment #39

Given there is little methylation data on this species to this point, it would be worth describing methylation location beyond just the identified markers (351-352). What are the main genomic locations of the 665,987 CCGG and 711,160 CCWGG sites?

We have now added a description of the genomic distribution of all methylated sites detected across samples (L181-182). 

Comment #40

It would be interesting to hear if any matching phenotypic data exists for this dataset. For example, for de novo methylation cluster 2 there are developmental terms. Could these arise from earlier / later stage samples? Were any of the de novo clusters predominantly populated by samples of a particular date? I know this would be hard to collect in retrospect but I think it would be an interesting observation.

Unfortunately, we do not have any detailed record about this. However, all the samples were collected between May and June 2015, and we selected individuals at very similar developmental stages. It is therefore unlikely that differences in developmental stage be responsible for major changes in DNA methylation patterns. But we cannot exclude that it may be responsible for a part of the observed differences. This has been clarified in the methods (L165-167) and discussion (593-598).

Minor points:

Comment #41

Line 23 – Given it’s a methylation paper I would avoid the use of imprint here.

The sentence was revised and the term imprint removed (L.18)

Comment #42

In line 83 it would be useful to have references to the stated stressors

Relevant references were added (L.91 , ref 5,18,27).

Comment #43

156 – The author says there was no reference genome available at the time. Are there plans to release the genome generated in this paper?

This was not mentioned in the manuscript, but was informed during the submission process and to the editor. The draft genome and the annotation will be released with the paper. Here is the information that has been added to the manuscript in the Data Availability section:

“All sequencing data generated in this study, including the raw sequencing reads (Illumina: SRR36185091; PacBio: SRR36185090) and the genome assembly are available from NCBI under Bioproject accession [PRJNA1368467]. A version of the genome assembly and annotation data (genome with rich annotation in various formats) is also available on Zenodo (10.5281/zenodo.17702558).” (L.683-687).

Comment #44

156 – I know it’s pedantic but please state high quality for first HC abbreviation.

We added the abbreviation after the first mention of high-quality reads: “758 million high-quality (HQ) reads were retained” (L.177).

Comment #46

Line 307 – It would be worth reiterating to the reader this is a second PCA and describing the difference between the two, especially given the confusion between 5/6 clusters given that this has 6 groups.

We clarify that this section refers to a second DAPC analysis using the six de novo groups (L. 278-279).

Comment #47

The study provides interesting results, and with clarification on the points above would make a suitable fit for the journal.

Thank you. We hope that our revisions have clarified all your concerns.

Reviewer 3 Report

Comments and Suggestions for Authors

The article title “Epigenetic Insights into Environmental Pressures in Natural Populations of a South American Mayfly” can be modified to represent the topic of this manuscript more adequately, I suggest: “Methylome Structure Based on Geographical Distribution of Natural Populations of a South American Mayfly.”

Simple summary.

Line 17: It is not clear yet if this is adaptive or not, but it is sufficiently clear that these epigenetic modifications are linked to stress responses. Please restrain adaptive speculation.

Both for Abstract and Intro.

Epigenetic mechanisms enable organisms to maintain homeostasis by regulating gene expression for essential cellular functions. while also allowing for adaptation and change in response to environmental stimuli.

Intro

Line 52:  Not only “selective”, but also ecological and evolutionary pressures in a broader sense.

Line  73: “Insects and plants” Methylations are quite present in microorganisms, such as bacteria, too. Fix this here, and cite. (Tourancheau et al., 2021).

Line 93: When presenting Limarí river, please add as a first figure a reference map showing in a small map of Chile where this watershed is located, and accompanied by a detailed map of Limarí with the collection points marked and a color code to represent the level of environmental stress to which each population/site is exposed. This can be used later as a base layer for overlapping your spatial graphs. This will contribute to clarity and precision for your audience.

Line 111: There is a problem with your hypothesis:

“We hypothesized that environmental stress is widespread in the Limarí watershed, leading to methylation patterns in A. torrens associated with biological processes to stress and regulation”.

As you are only evaluating the epigenetic marking response and there is no complementary data nor analysis of already published data provided about the local levels of stress (temperature, pollution, etc.) on the river sites considered to correlate with ecological epigenetic patterns. You earlier are providing references to state this, but your hypothesis is explicitly linking these two variables as the mechanistic explanation. Otherwise you are describing a geographical pattern of epigenetic marking, which is interesting, but is not reflecting your hypothesis goals.

This must be fixed by either 1) considering spatial correlation analysis and statistical models considering the matching of data from measurable variables of environmental stress evaluated in the watershed. This can be done with already published data to be pair with your current molecular evaluations.  2) Modifying the hypothesis to represent the evaluation of epigenetic marking in

2.Methods

My main methodological concern is already explained in the previous comment. If no evaluation of local stress variables is included, hypothesis must be adjusted.

Alternatively, please add a section on the mapping and analysis of river stress variables and how you are going to correlate or pair this with your epigenetic marking evaluation.

Regarding the presentation of the species, you must add the author and year of discovery of this species. Later in the text, you can use an abbreviation or a common name. In this case, in line 128, it should be: Andesiops torrens (Lugo-Ortiz & McCafferty, 1999) (Ephemeroptera: Baetidae).

When describing the model system, it is necessary to briefly comment on its life cycle and dispersion, as this factor may have a direct impact on the decisions made to test epigenetic regulation influences on this species. Epigenetic marking can be specifically triggered in a phase of insect development or maintained throughout the whole life cycle, even implying transgenerational consequences. All this is relevant to this manuscript, considering the idea of epigenetic molecular mechanisms as fast-paced mechanisms to cope with environmental stress. I recommend you to check and cite Fierro et al. 2025 work on Andesiops cycle and distribution (Fierro et al., 2025).

In the “Biological Sampling” section, cite Figure 1, which is recommended to add (a map with all sites marked in the Limarí watershed).

From the “Analysis of Methylome …” information, you can add a spatial evaluation of stress factors in the river and incorporate a suitable analysis that matches the epigenetic markings with the level of stress detected in the river. In this sense, stress evaluation values must be represented in an appendix table, citing the references if these are based on previous works’ data, and use this information to associate these levels of stress with epigenetic markings.

Alternatively, you can keep the method section as it is, but the hypothesis must be adjusted.

Results

These are primarily focused on describing the molecular results at the Limarí watershed sites and commenting on the biological processes involved in these changes. Once again, for testing the proposed hypothesis, there is a lack of explicit association with environmental stress.

Discussion

Line 442, it is not only unlikely in your study system, but as long as there is the chance to explore epimutations and ecological epigenetics, this is more the rule than the exception. Check (Richards et al., 2010, 2017; Liu et al., 2015; Burggren, 2017; Lamka et al., 2022).

In the discussion section, please distinguish between epigenetic regulations that are related to the conservation of housekeeping cellular functions (conservative regulation of processes) from  those that may suggest potential changes to cope with environmental stress (stress-related plasticity). To what extent is stress-related plasticity related to the development of epigenetically-triggered hormetic responses? (Vaiserman, 2011). Please discuss this further.

Line 536 Please discuss further the extent to which only considering nymphs in the samplings may affect the population pattern described. It would be desirable for authors to comment on the effect of dispersion behavior of this Ephemeroptera in the adult phase, considering its potential influence on the trends described (Fierro et al., 2025).

The article presents quite innovative and interesting results. I consider it is suitable for publication once these minor suggestions are addressed.

References cited:

Burggren, W. W. (2017). Epigenetics in insects: mechanisms, phenotypes and ecological and evolutionary implications. Advances in Insect Physiology 53, 1–30. doi: 10.1016/bs.aiip.2017.04.001

Fierro, P., Barrientos, P., Montiel, S., and Valdovinos, C. (2025). Distribution and life cycles of mayflies Andesiops torrens (Lugo-Ortiz and McCafferty, 1999) and Andesiops peruvianus (Ulmer, 1920) (Ephemeroptera: Baetidae): importance for environmental monitoring. Aquatic Insects 46, 38–54. doi: 10.1080/01650424.2024.2428863

Lamka, G. F., Harder, A. M., Sundaram, M., Schwartz, T. S., Christie, M. R., DeWoody, J. A., et al. (2022). Epigenetics in Ecology, Evolution, and Conservation. Frontiers in Ecology and Evolution 10. Available at: https://www.frontiersin.org/articles/10.3389/fevo.2022.871791 (Accessed July 13, 2023).

Liu, S., Sun, K., Jiang, T., and Feng, J. (2015). Natural epigenetic variation in bats and its role in evolution. The Journal of experimental biology 218, 100–106. doi: 10.1242/jeb.107243

Richards, C. L., Alonso, C., Becker, C., Bossdorf, O., Bucher, E., Colomé-Tatché, M., et al. (2017). Ecological plant epigenetics: Evidence from model and non-model species, and the way forward. Ecology Letters 20, 1576–1590. doi: 10.1111/ele.12858

Richards, C. L., Bossdorf, O., and Verhoeven, K. J. F. (2010). Understanding natural epigenetic variation. The New Phytologist 187, 562–564. Available at: https://www.jstor.org/stable/40792404 (Accessed June 1, 2023).

Tourancheau, A., Mead, E. A., Zhang, X.-S., and Fang, G. (2021). Discovering multiple types of DNA methylation from bacteria and microbiome using nanopore sequencing. Nat Methods 18, 491–498. doi: 10.1038/s41592-021-01109-3

Vaiserman, A. M. (2011). Hormesis and epigenetics: Is there a link? Ageing Research Reviews 10, 413–421. doi: 10.1016/j.arr.2011.01.004

Same comments sent to Editors

Author Response

Reviewer 3:
Comments and Suggestions for Authors

Comment #48

The article title “Epigenetic Insights into Environmental Pressures in Natural Populations of a South American Mayfly” can be modified to represent the topic of this manuscript more adequately, I suggest: “Methylome Structure Based on Geographical Distribution of Natural Populations of a South American Mayfly.”

We revised the title to more accurately reflect the scope and content of the manuscript. The new title is: “Patterns and functional insights of DNA methylation variation in a South American mayfly across an agriculturally impacted semi-arid watershed.”

Comment #49

Simple summary.

Line 17: It is not clear yet if this is adaptive or not, but it is sufficiently clear that these epigenetic modifications are linked to stress responses. Please restrain adaptive speculation.

We revised the section to remove adaptive wording, such as “adapt,” “imprint,” and “helping them survive”, and now refer instead to the pressures faced by natural populations. (L.20-22, 25, 28).

Comment #50

Both for Abstract and Intro.

Epigenetic mechanisms enable organisms to maintain homeostasis by regulating gene expression for essential cellular functions, while also allowing for adaptation and change in response to environmental stimuli.

We incorporated this idea into both the Abstract and Introduction. The revised text now highlights the dual role of epigenetic mechanisms in maintaining homeostasis and enabling responses to environmental stimuli (Abstract L.33-35; Introduction L65-68).

Comment #51

Intro

Line 52:  Not only “selective”, but also ecological and evolutionary pressures in a broader sense.

We agree with the reviewer and revised the sentence accordingly (L. 56).

Comment #52

Line  73: “Insects and plants” Methylations are quite present in microorganisms, such as bacteria, too. Fix this here, and cite. (Tourancheau et al., 2021).

We revised the sentence to include microorganisms and added the suggested citation (L.79-80).

Comment #53

Line 93: When presenting Limarí river, please add as a first figure a reference map showing in a small map of Chile where this watershed is located, and accompanied by a detailed map of Limarí with the collection points marked and a color code to represent the level of environmental stress to which each population/site is exposed. This can be used later as a base layer for overlapping your spatial graphs. This will contribute to clarity and precision for your audience.

We have included a map with the vegetation cover (i.e. mainly agriculture) and reported pesticide contamination along the watershed. Evaluating environmental stress and color-coding it is no easy task, especially in lotic ecosystems. According to the river continuum concept, water condition generally degrades downstream. But a precise characterization of the water condition would also require a longitudinal monitoring, which could not be done in this project. How to characterize environmental stress is another challenge. In addition, based on recent publications, pesticide contamination can be high in upstream sites where agriculture is not so intense (Gouin et al., 2019, 2025), so there is no correlation for example between the extent of agricultural land use and level of pesticide contamination. We hope that the map we provide helps better understand the context of the study.

Comment #54

Line 111: There is a problem with your hypothesis:

“We hypothesized that environmental stress is widespread in the Limarí watershed, leading to methylation patterns in A. torrens associated with biological processes to stress and regulation”.

As you are only evaluating the epigenetic marking response and there is no complementary data nor analysis of already published data provided about the local levels of stress (temperature, pollution, etc.) on the river sites considered to correlate with ecological epigenetic patterns. You earlier are providing references to state this, but your hypothesis is explicitly linking these two variables as the mechanistic explanation. Otherwise you are describing a geographical pattern of epigenetic marking, which is interesting, but is not reflecting your hypothesis goals.

This must be fixed by either 1) considering spatial correlation analysis and statistical models considering the matching of data from measurable variables of environmental stress evaluated in the watershed. This can be done with already published data to be pair with your current molecular evaluations.  2) Modifying the hypothesis to represent the evaluation of epigenetic marking.

We agree that our original hypothesis implied a mechanistic link with environmental stress that was not directly tested with environmental measurements in this study. As explained in our response to Comment #1, we opted not to conduct new spatial or environmental correlation analyses, because these could only be meaningfully applied to geographic site-based groupings and not to the de novo methylation clusters, which were a central focus of our study. To address this concern, we revised the hypothesis to avoid implying a direct mechanistic link with stress and to more accurately reflect the scope of our analyses (L. 121-124).

Comment #55

2.Methods

My main methodological concern is already explained in the previous comment. If no evaluation of local stress variables is included, hypothesis must be adjusted.

Alternatively, please add a section on the mapping and analysis of river stress variables and how you are going to correlate or pair this with your epigenetic marking evaluation.

As noted in our response to Comment #1 and 54, we did not incorporate additional environmental analyses but revised the hypothesis.

Comment #56

Regarding the presentation of the species, you must add the author and year of discovery of this species. Later in the text, you can use an abbreviation or a common name. In this case, in line 128, it should be: Andesiops torrens (Lugo-Ortiz & McCafferty, 1999) (Ephemeroptera: Baetidae).

We added the author and year of description for Andesiops torrens as suggested (L.98)

Comment #57

When describing the model system, it is necessary to briefly comment on its life cycle and dispersion, as this factor may have a direct impact on the decisions made to test epigenetic regulation influences on this species. Epigenetic marking can be specifically triggered in a phase of insect development or maintained throughout the whole life cycle, even implying transgenerational consequences. All this is relevant to this manuscript, considering the idea of epigenetic molecular mechanisms as fast-paced mechanisms to cope with environmental stress. I recommend you to check and cite Fierro et al. 2025 work on Andesiops cycle and distribution (Fierro et al., 2025).

We have added the following information in the manuscript Andesiops torrens is a species endemic to Chile, distributed from the Limarí watershed in the north to the southern tip of Patagonia (Fierro et al., 2024). We also explain that this species exhibits a univoltine life cycle, with emergence occurring between August to October in central Chile (Fierro et al., 2024).” L.136-139.

Comment #58
In the “Biological Sampling” section, cite Figure 1, which is recommended to add (a map with all sites marked in the Limarí watershed).

                Done (L. 163).

Comment #59

From the “Analysis of Methylome …” information, you can add a spatial evaluation of stress factors in the river and incorporate a suitable analysis that matches the epigenetic markings with the level of stress detected in the river. In this sense, stress evaluation values must be represented in an appendix table, citing the references if these are based on previous works’ data, and use this information to associate these levels of stress with epigenetic markings.

 Alternatively, you can keep the method section as it is, but the hypothesis must be adjusted.

See our responses to comments #2 and #54.

Comment #60

Results

These are primarily focused on describing the molecular results at the Limarí watershed sites and commenting on the biological processes involved in these changes. Once again, for testing the proposed hypothesis, there is a lack of explicit association with environmental stress.

We agree. We revised the relevant sections of the manuscript to ensure that the results are coherent with the revised scope and goals of the study.

Comment #61

Discussion

 Line 442, it is not only unlikely in your study system, but as long as there is the chance to explore epimutations and ecological epigenetics, this is more the rule than the exception. Check (Richards et al., 2010, 2017; Liu et al., 2015; Burggren, 2017; Lamka et al., 2022).

We revised the paragraph to highlight that genetically independent epigenetic variation could be common and incorporated relevant references (L.469-472).

Comment #62

In the discussion section, please distinguish between epigenetic regulations that are related to the conservation of housekeeping cellular functions (conservative regulation of processes) from  those that may suggest potential changes to cope with environmental stress (stress-related plasticity). To what extent is stress-related plasticity related to the development of epigenetically-triggered hormetic responses? (Vaiserman, 2011). Please discuss this further.

We thank the reviewer for this valuable suggestion. In the revised version, we now explicitly distinguish between methylation patterns associated with housekeeping, core cellular maintenance and those corresponding to environmentally induced stress-related plasticity. In addition, following the reviewer’s recommendation, we added a concise discussion of hormesis and the potential role of epigenetic regulation in mediating hormetic-like responses under recurrent or chronic stress exposure. This new paragraph (Discussion, 613-629) explains the relevance of hormesis to our system, integrates Vaiserman (2011), and interprets our functional enrichment patterns (particularly in Groups D, E, and B) in this context while remaining cautious about causal inference. The revised text emphasizes that although hormesis cannot be demonstrated directly with our data, the coexistence of housekeeping-related and stress-induced methylation profiles suggests that A. torrens may express a spectrum of epigenetically mediated responses to persistent environmental pressures.

Comment #63

Line 536 Please discuss further the extent to which only considering nymphs in the samplings may affect the population pattern described. It would be desirable for authors to comment on the effect of dispersion behavior of this Ephemeroptera in the adult phase, considering its potential influence on the trends described (Fierro et al., 2025).

This an interesting comment. We have added information in the discussion using the Gouin et al. (2019) paper revealing a connected metacommunity in this watershed, and discussed how downstream drift of aquatic nymphs and dispersal of adults can affect the patterns of DNA methylation (L.579-584). We also included a reference of Almudi et al. (2020) to specify that gene expression patterns are stage-specific in mayflies from the Baetidae family (L.593-594), and give more arguments that our sampling strategy is unlikely to have produced strong effects on the variation of our DNA methylation profiles.

Comment #64

The article presents quite innovative and interesting results. I consider it is suitable for publication once these minor suggestions are addressed.

Thank you. We hope that the revisions made throughout the manuscript adequately address all your comments and concerns.

Comment #65

We added various of the references suggested by the reviewer to the new version of the manuscript.

References cited:

Burggren, W. W. (2017). Epigenetics in insects: mechanisms, phenotypes and ecological and evolutionary implications. Advances in Insect Physiology 53, 1–30. doi: 10.1016/bs.aiip.2017.04.001

Fierro, P., Barrientos, P., Montiel, S., and Valdovinos, C. (2025). Distribution and life cycles of mayflies Andesiops torrens (Lugo-Ortiz and McCafferty, 1999) and Andesiops peruvianus (Ulmer, 1920) (Ephemeroptera: Baetidae): importance for environmental monitoring. Aquatic Insects 46, 38–54. doi: 10.1080/01650424.2024.2428863

Lamka, G. F., Harder, A. M., Sundaram, M., Schwartz, T. S., Christie, M. R., DeWoody, J. A., et al. (2022). Epigenetics in Ecology, Evolution, and Conservation. Frontiers in Ecology and Evolution 10. Available at: https://www.frontiersin.org/articles/10.3389/fevo.2022.871791 (Accessed July 13, 2023).

Liu, S., Sun, K., Jiang, T., and Feng, J. (2015). Natural epigenetic variation in bats and its role in evolution. The Journal of experimental biology 218, 100–106. doi: 10.1242/jeb.107243

Richards, C. L., Alonso, C., Becker, C., Bossdorf, O., Bucher, E., Colomé-Tatché, M., et al. (2017). Ecological plant epigenetics: Evidence from model and non-model species, and the way forward. Ecology Letters 20, 1576–1590. doi: 10.1111/ele.12858

Richards, C. L., Bossdorf, O., and Verhoeven, K. J. F. (2010). Understanding natural epigenetic variation. The New Phytologist 187, 562–564. Available at: https://www.jstor.org/stable/40792404 (Accessed June 1, 2023).

Tourancheau, A., Mead, E. A., Zhang, X.-S., and Fang, G. (2021). Discovering multiple types of DNA methylation from bacteria and microbiome using nanopore sequencing. Nat Methods 18, 491–498. doi: 10.1038/s41592-021-01109-3

Vaiserman, A. M. (2011). Hormesis and epigenetics: Is there a link? Ageing Research Reviews 10, 413–421. doi: 10.1016/j.arr.2011.01.004

Round 2

Reviewer 1 Report

Comments and Suggestions for Authors

Most of my comments were considered in the revision, and the manuscript is clearly in a better and more coherent state, both regarding text and figures. The supplementary figures are readable now. The emphasis on environment has also been lessened. I believe the manuscript can be accepted as is, however, see suggestion below.

I do, however, have a suggestion to ease interpretation of the results: The authors could consider making figure panels for each grouping (denovo and site based). That may give easier access to interpretation of the groups and functional annotations, instead of having to flip back and fourth between figures.
Consider this suggestion, as good figures (or panels) are key for readers.

Author Response

Comment#1:
Most of my comments were considered in the revision, and the manuscript is clearly in a better and more coherent state, both regarding text and figures. The supplementary figures are readable now. The emphasis on environment has also been lessened. I believe the manuscript can be accepted as is, however, see suggestion below.

I do, however, have a suggestion to ease interpretation of the results: The authors could consider making figure panels for each grouping (denovo and site based). That may give easier access to interpretation of the groups and functional annotations, instead of having to flip back and fourth between figures.
Consider this suggestion, as good figures (or panels) are key for readers.

Response to Comment #1
Following the reviewer’s recommendation, we have reorganized the relevant figures into panel formats that combine clustering outputs with their functional profiles:

  • Figure 2 is now a panel figure that integrates:
    (a) population clustering based on DAPC site scores,
    (b) the corresponding geographical locations of sampling sites (with ID numbers matching Table S1), and
    (c) the functional profiles of the population-based methylation groups (A–E), shown as colored arcs representing the major enriched biological processes detected in the GO analyses.

  • Figure 4 has similarly been revised into a unified panel that includes:
    (a) the geographic distribution of the six de novo methylation groups across sampling sites (as pie charts), and
    (b) the associated functional profiles of the de novo groups (1–6), with colored arcs summarizing the dominant biological processes enriched in each group.